# Highly robust and soft biohybrid mechanoluminescence for optical signaling and illumination

Chenghai Li[1], Qiguang He[1], Yang Wang[2], Zhijian Wang[1], Zijun Wang[2], Raja Annapooranan [2], Michael I. Latz [3] & Shengqiang Cai [1,2✉]

Biohybrid is a newly emerging and promising approach to construct soft robotics and soft machines with novel functions, high energy efficiency, great adaptivity and intelligence. Despite many unique advantages of biohybrid systems, it is well known that most biohybrid systems have a relatively short lifetime, require complex fabrication process, and only remain functional with careful maintenance. Herein, we introduce a simple method to create a highly robust and power-free soft biohybrid mechanoluminescence, by encapsulating dinoflagellates, bioluminescent unicellular marine algae, into soft elastomeric chambers. The dinoflagellates retain their intrinsic bioluminescence, which is a near-instantaneous light response to mechanical forces. We demonstrate the robustness of various geometries of biohybrid mechanoluminescent devices, as well as potential applications such as visualizing external mechanical perturbations, deformation-induced illumination, and optical signaling in a dark environment. Our biohybrid mechanoluminescent devices are ultra-sensitive with fast response time and can maintain their light emission capability for weeks without special maintenance.

[1] Department of Mechanical and Aerospace Engineering, University of California, San Diego, La Jolla, CA 92093, USA. [2] Materials Science and Engineering Program, University of California, San Diego, La Jolla, CA 92093, USA. [3] Scripps Institution of Oceanography, University of California, San Diego, La Jolla, CA 92093, USA. ✉email: shqcai@ucsd.edu

The biohybrid approach has been recently intensively explored to construct soft devices and robotics of different forms with diverse functionalities[1–4]. For example, muscular cells such as cardiomyocytes have been integrated with elastomeric or hydrogel structures to achieve customizable and complex actuations with low power consumption[5,6]. Genetically engineered *E. coli* has been integrated with soft grippers or hydrogels, sensing specific chemical stimuli and then generating fluorescence[7–10]. Neurons have also been explored to autonomously control the actuation of a biohybrid swimmer[11]. By taking advantage of both customized biological functions and engineering materials/structures, biohybrid systems may reproduce some characteristics of biology including sophisticated performance, intelligent and autonomous control, high energy efficiency, and high adaptivity to the surroundings[1–4]. Though biohybrid approach has opened a new horizon in the design of soft robots and soft machines, most biohybrid systems have relatively short lifetimes, typically within several days, and require special maintenance to maintain their normal functionality and prolong their lifetimes[3,7–10,12]. In addition, the time scale of the response of many biohybrid systems for sensing external stimuli is typically long, e.g., several hours for *E. coli* based biohybrid sensors[7–9], a limiting factor for many applications.

The capability of visualizing mechanical deformations or forces has been recently introduced into various soft robotic devices for proprioceptive and exteroceptive sensing[13–15]. For instance, soft elastomers that can change colors by deformation, known as mechanochromism, have been used to build a soft gripper to visualize the extent of deformation[14]. A similar mechanochromic response has also been achieved in liquid crystal elastomer-based soft actuators[15]. Moreover, photonic crystal arrays have been explored to visualize mechanical deformations/forces[16–18]. However, none of these designs can emit light, so it is still very challenging to visualize deformations or forces in a dark environment. To produce light, a highly stretchable electroluminescent skin has been recently developed for soft robots for both optical signaling and tactile sensing[13]. Nevertheless, such a design requires electrical components and power supplies, which may complicate the system and the fabrication of soft robots.

Mechanoluminescence is a process of light emission caused by mechanical actions on a material, which can be easily detected in a dark environment[19–21]. Most mechanoluminescent materials are inorganic solids such as $SrAl_2O_4:Eu^{2+}$ and $ZnS:Cu/Mn^{2+}$, which are stiff and brittle, and thus not compatible with soft and stretchable devices[19–21]. Recently, mechanoluminescent ceramic particles have been embedded into elastomers to make a stretchable composite, which however still requires a large strain rate and stress to produce light[19,22,23]. More recently, mechanophore molecules have also been incorporated into polymer networks to make soft and stretchable elastomers, most of which can emit light caused by bond breakage[24,25]. However, the destructive feature does not allow the reuse of the material and the intensity of the generated light is usually very weak. In contrast, bioluminescent phenomena are quite common in nature, generating light with high energy efficiency[26–28]. Designing an electronics-free and soft biohybrid mechanoluminescent device with simple fabrication process, instantaneous response, high sensitivity, and a long lifetime may enable untethered, small scale, and intelligent systems. However, to our knowledge, such designs have not yet been achieved.

The goal of the present study is to explore the feasibility of building a soft and robust biohybrid mechanoluminescent device that incorporates bioluminescent dinoflagellates, unicellular marine algae that are common sources of ocean bioluminescence[29,30]. Bioluminescent organisms are broadly distributed across the marine and terrestrial environments, with their light emission functioning in defense against predators, mate attraction, prey attraction and illumination, and intraspecific communication[26–28]. Here, we present a simple approach to integrate bioluminescent dinoflagellates with soft materials to build a highly stretchable, power-free, and ultra-sensitive mechanoluminescence. Dinoflagellate bioluminescence is a response to mechanical stress[31–33], due to flow and touch, involving a complex intracellular signaling pathway[31–36]. Such bioluminescence is a visibly remarkable phenomenon often occurring in coastal areas (Supplementary Fig. 1A). Stress levels as low as several Pa can activate dinoflagellate bioluminescence with a latency from stimulus to response of only 15–20 ms[37–41], resulting in ultra-high and near-instantaneous force sensitivity. Most notably, dinoflagellates can be very robust, tolerating different conditions with a functional bioluminescent response that serves as a reporter of mechanical stress for ocean[42,43] and laboratory[44–47] visualization (Supplementary Movie 1).

In this work, our study includes: (1) Envision the potential applications of the soft biohybrid mechanoluminescence developed in the current study including visualizing mechanical perturbations, deformation-induced illumination, and optical signaling in a dark environment (Fig. 1). (2) Illustrate the fabrication method of the biohybrid mechanoluminescent devices, demonstrate the working principle and extend the color range. (3) Demonstrate how the design of the elastomeric structure can greatly affect the light intensity of the biohybrid device due to the flow condition changes. (4) Explore simple integrations of such devices with soft actuators/robotics and demonstrate their potential applications in the dark. First, the mechanoluminescent devices can be used to visualize external mechanical perturbations caused by direct contact or air flow. Second, the deformation of the soft device may enable the illumination of the surrounding dark environment. Third, by designing the deformation mode of the soft actuator/robot, diverse mechanoluminescent patterns can be displayed for potential optical signaling in the dark. Finally, we design an untethered magnetically controlled soft robot with a closed system. Most importantly, our untethered biohybrid mechanoluminescent robot is ultra-sensitive with fast response that can maintain its light emission capability for weeks without special maintenance. We hope such integrations of the bioluminescence and engineering soft materials/structures will promote more intelligent and functional applications.

## Results

**Potential applications of the biohybrid mechanoluminescence.** We propose that the soft biohybrid mechanoluminescence constructed in the current study may find its potential applications in soft robotic devices (Fig. 1). In the light phase, the soft robot integrated with dinoflagellate culture solution is charged with light for photosynthesis to produce oxygen, providing energy for the organism. In the dark phase and in a dark environment, both internal and external stimuli can deform the soft robot, which induces the flow of internal fluid and thus activates the bioluminescence for potential applications in several aspects. First, the mechanoluminescent soft robot skin can be used to visualize external mechanical perturbations including contact stimuli (e.g., compression) and contactless stimuli (e.g., air flow). Second, the active actuation-induced and passive disturbance-induced mechanoluminescence of the biohybrid soft robot can illuminate the surrounding dark environment to be visible. Third, by designing the deformation mode of the soft actuator/robot, diverse mechanoluminescent patterns can be displayed for potential optical signaling in the dark. We envision that the facile incorporation of such electronics-free and biohybrid mechanoluminescence into soft robotics may enable untethered, small

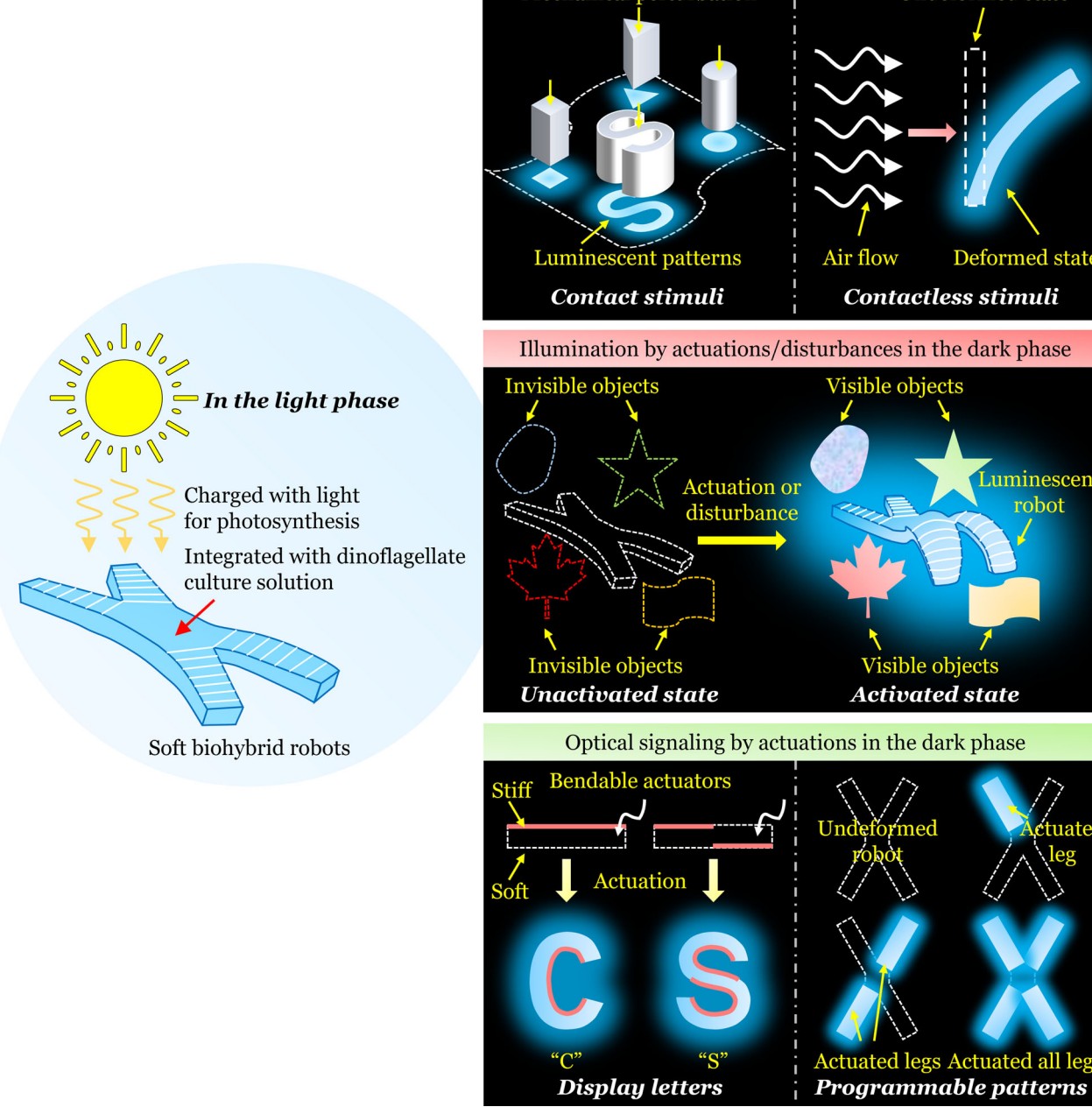

**Fig. 1 Schematic of the working principle and potential applications of the biohybrid mechanoluminescence.** In the light phase, the soft biohybrid robot integrated with dinoflagellate culture solution is charged with sunlight for photosynthesis to produce oxygen, providing energy for the organism. In the dark phase, the mechanically induced bioluminescence of the soft biohybrid robot can visualize mechanical perturbations, illuminate surrounding area, and produce optical signals.

scale, and intelligent systems with instantaneous response, high sensitivity, and a long lifetime without special maintenance.

**Design of the soft biohybrid mechanoluminescence.** To fabricate a biohybrid mechanoluminescent device, we encapsulated a culture solution of the photosynthetic dinoflagellate *Pyrocystis lunula* (*P. lunula*) (Supplementary Fig. 1B) by a soft and transparent elastomer (PDMS) chamber (Fig. 2A) to create a biohybrid device (see Supplementary Fig. 2 for details). This species was selected because it is widespread globally in many oceans of the world[48–52], indicating that it is tolerant to a broad range of

environmental conditions. In addition, dinoflagellates of the genus *Pyrocystis* are known to tolerate being enclosed within containers[53]. Note that PDMS is highly permeable to gas exchange[8,54–56], which guarantees the survival of enclosed bacteria/cells. During the dark phase of the dinoflagellate maintenance cycle, when bioluminescence is expressed, deformation of the biohybrid device (e.g., by stretching, twisting, bending, compression, vibration, impact, and other stimuli) caused motion of the encapsulated fluid, resulting in a shear stress applied to the dinoflagellates to activate their bioluminescence. The biohybrid device is highly transparent to visible light (Fig. 2B, Left and Supplementary Fig. 3), allowing both the transmission of

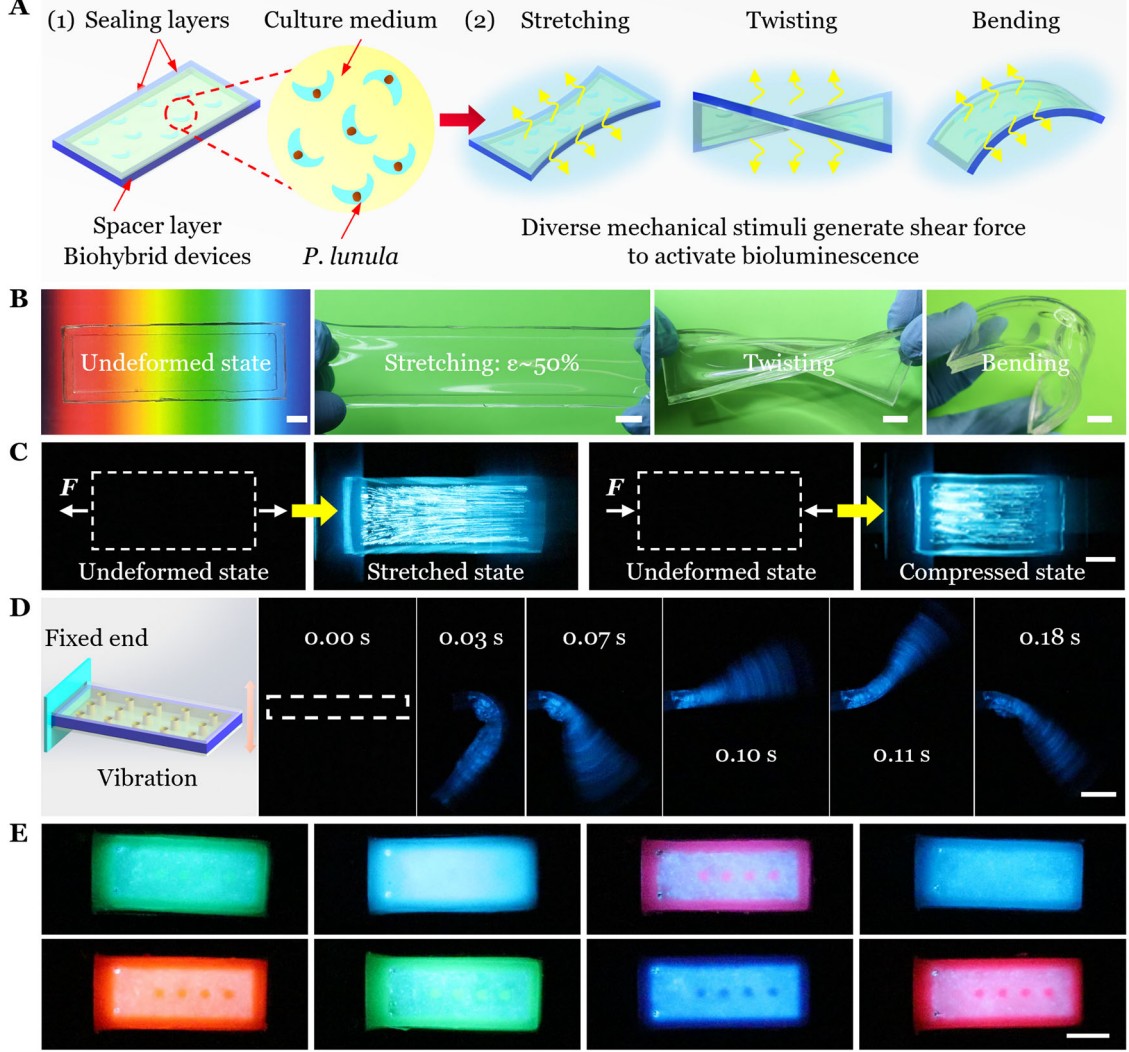

**Fig. 2 Design, working principle and simple demonstrations of the soft biohybrid mechanoluminescence. A** The schematic shows the typical design of a biohybrid mechanoluminescent device. Under different external mechanical stimuli (e.g., stretching, twisting, bending, compression, impact, and others), the deformation of the soft device induces the flow of internal fluid and thus activates the bioluminescence. **B** The biohybrid device is highly transparent and can be stretched, twisted, or bent without failure. Scale bar, 1 cm. **C** Bioluminescence of the biohybrid device under stretched and compressed states. Images are long time exposures (specified in Methods). Scale bar, 1 cm. **D** Side view of bioluminescence within the biohybrid device during its free vibration with the left end being fixed. Elapsed times are given in each frame. Scale bar, 1 cm. **E** Top view of bioluminescence due to free vibration when the elastomer matrix is mixed with dyes of certain colors, generating light of different colors. Scale bar, 1 cm.

illumination for the dinoflagellates to carry out photosynthesis to produce oxygen during the light phase, and the viewing of intrinsic bioluminescence during the dark phase. Because the elastomer is soft and highly deformable, the biohybrid device can be greatly stretched, twisted and bended without rupture (Fig. 2B). Specifically, the results of uniaxial tension tests showed that the biohybrid device has a large stretchability $\lambda_c$ around 1.9 (Supplementary Fig. 4).

We next demonstrate the mechanoluminescence of the biohybrid device under simple stretching and compression loadings. In the undeformed state, the device was totally dark and invisible in a dark environment. While in the deformed state, the flow of internal fluid activated the bioluminescence and thus the device emitted light (Fig. 2C), which was imaged by a sensitive digital camera (specified in Methods).

To magnify the shear stress in the solution generated by the deformation of the biohybrid device, we further designed pillars on the bottom surface of the chamber (Supplementary Fig. 5). During the dark phase, the biohybrid device was fixed at one end

and an instantaneous load was applied at the tip of the other end, causing the free vibration of the cantilever beam (Fig. 2D, Left). Bioluminescence stimulated within the cantilever beam was imaged (Fig. 2D and Supplementary Movie 2) and quantified as a function of the pillar height $h$ (see next section).

The light emission of most bioluminescent organisms including dinoflagellates has a spectral maximum in the blue-green, optimized for propagation through water[26,34,53,57]. However, practical applications may require bioluminescent light of different colors. By simply mixing color dyes (~1 wt%) with the precursor of the elastomeric cantilever beam (Supplementary Fig. 6), we demonstrated that the original blue-green bioluminescence can be shifted to other colors (Fig. 2E and Supplementary Movie 3).

**Characterizations of the soft biohybrid mechanoluminescent devices.** Next, we quantified how the design of the elastomer chamber and the resultant fluid flow under mechanical loadings affected the bioluminescence intensity of the biohybrid devices.

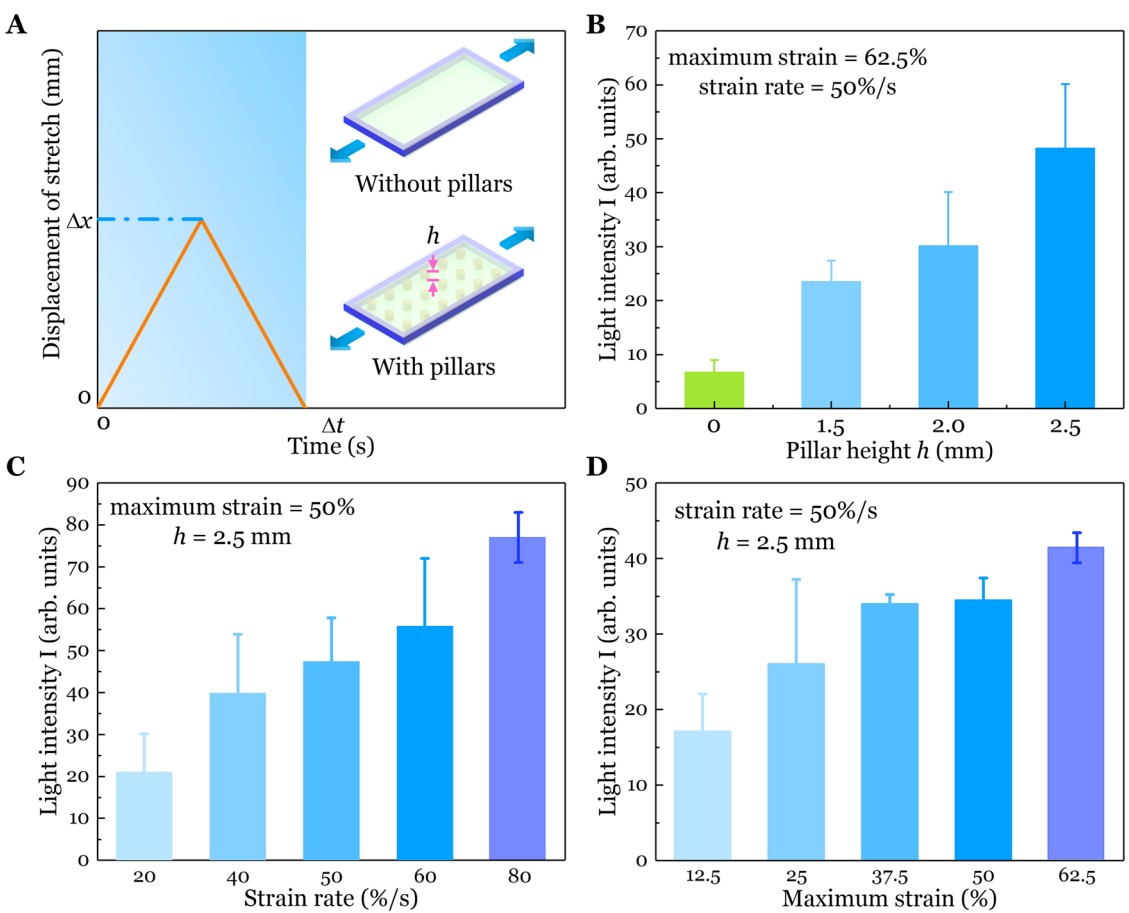

**Fig. 3 Characterizations of the light intensity of biohybrid devices under various loading conditions. A** The inset shows two kinds of chamber designs: "Without pillars" possesses a flat surface for the bottom layers while "With pillars" contains cylindrical pillars with the height denoted by $h$ to the bottom surface of the chamber. Chamber volume is kept the same for all configurations. We applied one cycle of the triangle loading-unloading test to the devices with the maximum displacement as $\Delta x$ and period as $\Delta t$. Light emission was derived from camera imaging of stimulated bioluminescence (see Methods for details). **B** Light intensity based on grayscale values of the stimulated bioluminescence increased as a function of the height of the pillars, with "$h = 0$ mm" referring to the chamber design without pillars. **C** Light intensity as a function of strain rate for a fixed maximum strain of 50% for chambers with $h = 2.5$ mm pillars. **D** Light intensity as a function of maximum strain for a fixed strain rate of 50% s$^{-1}$ for chambers with $h = 2.5$ mm pillars. Values in (**B**), (**C**), and (**D**) represent averages with standard deviations for $N = 3$ replicates.

We fabricated two similar elastomer chambers (Fig. 3A, Insert). For the first design, the chamber had flat and smooth inner surfaces (Supplementary Fig. 4A), while for the second design, cylindrical pillars were fabricated on the bottom surface of the chamber (Supplementary Fig. 5). During the dark phase, we applied one cycle of the triangle loading-unloading test to the device in a dark environment with the maximum displacement as $\Delta x$ and the period as $\Delta t$. Simultaneously, we imaged light emission from the biohybrid device using photographs made by the digital camera with an exposure time > $\Delta t$ (specified in Methods). We then calculated the light intensity based on grayscale values in arbitrary units (arb. units) determined from MATLAB image analysis of the color images (Supplementary Fig. 7, see "Methods" for details).

We first examined the correlation between the concentration of dinoflagellate cells and measured light intensity from the images of bioluminescence (Supplementary Fig. 8 and Supplementary Note 1). Results showed that the light intensity generally increased with the concentration of cells (Supplementary Fig. 8D).

We then studied the effect of the pillar height $h$ on light intensity, keeping the total volume of the culture solution the same, for a maximum strain and strain rate of 62.5 and 50% s$^{-1}$, respectively (Fig. 3B, Supplementary Fig. 5D and 9). Light

intensity increased with pillar height $h$, which can be attributed to enhanced fluid motion with taller pillars.

Next, for a maximum strain of 50%, we studied the effect of strain rate on light intensity for the chamber with $h = 2.5$ mm pillars from 20% s$^{-1}$ to 80% s$^{-1}$ (Fig. 3C and Supplementary Fig. 10A). Results showed that the light intensity increased with the increase of strain rate. We also studied the effect of the maximal strain on light intensity for the chamber with $h = 2.5$ mm pillars under a fixed strain rate of 50% s$^{-1}$ (Fig. 3D and Supplementary Fig. 10B). Results showed that the light intensity increased with maximum strain from 12.5% to 62.5%. We also conducted the same experiments on the chambers without pillars and obtained similar results (Supplementary Fig. 11). Under the same loading conditions, the chamber with pillars generated higher light intensity than the chamber without pillars. These experimental results are consistent with previous studies of dinoflagellate bioluminescence showing that higher loading strain rate or larger deformation generates higher shear force and thus a higher intensity of stimulated bioluminescence[32,33,58–60].

Next, we studied the effect of cyclic loading-unloading stretching and compression on light intensity using the chamber with $h = 2.5$ mm pillars (Supplementary Fig. 12 and Supplementary Note 2). Light intensity decreased exponentially with cycle

number for both stretching (Supplementary Fig. 12A, C) and compression (Supplementary Fig. 12B, D). We also studied the recovery ability of bioluminescence of the biohybrid devices (Supplementary Fig. 13 and Supplementary Note 3). Results showed that full recovery of bioluminescence occurred after 30 min (Supplementary Fig. 13B). Finally, we characterized the viability of the biohybrid devices by measuring the light intensity under the same loading and also measuring the concentration of cells inside the devices for 15 continuous days (Supplementary Fig. 14 and Supplementary Note 4). Our biohybrid devices could maintain a high relative light intensity (>55% of initial values) for at least 15 days (Supplementary Fig. 14B), indicating adequate oxygen and carbon dioxide supply for the dinoflagellates and also high viability.

**Visualizing external mechanical perturbations in the dark**. In this section, we demonstrate that the mechanoluminescent device may be used as soft robot skin or its component to visualize external mechanical perturbations including both contact and contactless stimuli in the dark.

First, we explored the use of the soft biohybrid mechanoluminescence for sensing dynamic contact. We manually manipulated 3D printed stiff plastic indenters of different shapes (e.g., cylinder, rectangular block, triangular prism, and cone) to gradually compress and then release from the biohybrid device (Fig. 4A–D and Supplementary Fig. 15), and simultaneously recorded the stimulated bioluminescence (Supplementary Movie 4). The sequential luminescent patterns viewed from underneath the device corresponded well to the geometry of the objects (Fig. 4A–D). Bioluminescence increased during the compression phase, and then decreased during the release phase.

Such a device serves as a visible mechanoluminescent sensor for the dynamic contact in a dark environment. By monitoring the evolution of luminescent patterns, we could qualitatively extract the information of the object geometry and the dynamic contact. Each dinoflagellate cell serves as a sensing unit providing the ultra-high spatial resolution. And the low shear stress (~Pa to ~kPa) needed to activate the bioluminescence offers ultra-high force sensitivity. Quantitative correlations between the distribution of compression stress and lightening patterns may be established to accurately reflect the object geometry and the dynamic process in future studies.

We next demonstrate the mechanoluminescent behaviors of the biohybrid device for visualizing the trace of local compression. We fabricated a multichambered panel with $4 \times 4$ internal chambers (Supplementary Fig. 16) and then infused the chambers with the dinoflagellate culture solution. We used a stylus to write the letters "UCSD" on the surface during the dark phase (Fig. 4E, Left). During the writing, the stylus compressed the solution underneath, activating the bioluminescence of dinoflagellates, which produced light in the form of the letters (Fig. 4E). Based on the high sensitivity, fast response, and power-free feature, our biohybrid panel may be suitable to record both the trace of compression and the local applied pressure with further quantitative analysis, enabling potential real-time pressure mapping visible sensors.

We also utilized the biohybrid device to detect contactless stimulation by a mild air flow. We fabricated multiple colored cantilevers with a slender shape (Supplementary Fig. 17) and then fixed them to a base for the air flow test (Fig. 4F, Left). Under dark conditions with no air flow, no bioluminescence emitted from the device (Fig. 4F, Middle). Then, an air flow with a rate of ~45 L min⁻¹ bent and deformed the cantilevers and thus stimulated bioluminescence within the device, resulting in light emission from the colored biohybrid device (Fig. 4F, Right). Such

devices may have potential applications in harvesting environmental air flow for illumination or display functions in the dark environment as a sustainable and environmentally friendly option.

**Illumination by actuations or mechanical disturbances in the dark**. In this section, we demonstrate that in the dark environment the biohybrid mechanoluminescent soft actuator/robot can emit light to make the surrounding area visible either by active actuations or external disturbances.

We firstly designed a hydraulic actuated soft crawling robot (Fig. 5A and Supplementary Fig. 18) and then infused it with dinoflagellate culture solution. A piece of acrylic plate was attached to the front of the robot to induce friction difference on the substrate. In the dark environment, we placed the robot on an incline with the angle of 17°. Then, a syringe reservoir of dinoflagellate culture solution was used to actuate the robot. Before actuation, the undeformed robot did not emit any light (Fig. 5B, Left). Upon active hydraulic actuation, the soft robot inflated and then crawled forward due to the friction difference on the incline and simultaneously emitted light to illuminate the surrounding background (Fig. 5B and Supplementary Movie 5).

We further demonstrate that external disturbance can also induce the illumination of the soft robot without requiring energy input from itself. We designed an untethered tetrapod-like soft robot with four legs and infused the four legs with dinoflagellate culture solution (Fig. 5C and Supplementary Fig. 19). In the dark environment, before any external disturbance, the robot was totally dark (Fig. 5D, Left). Then, we manually pressed each leg using a finger. As a result, the disturbed leg emitted the brightest bioluminescence and simultaneously illuminated the surrounding background (Fig. 5D and Supplementary Movie 5).

**Optical signaling by actuations in the dark**. In this section, we demonstrate that by triggering different actuation modes of the soft actuator/robot, diverse mechanoluminescent patterns can be displayed for potential optical signaling in the dark.

We firstly designed a soft and hydraulically actuated bending actuator (Fig. 6A and Supplementary Fig. 20A, C). The unidirectional bending actuator was composed of a soft inflatable part and a stiff constraining layer (Fig. 6B, Left). Before the actuation, the undeformed actuator was invisible in the dark. Upon hydraulic actuation, the soft actuator bent to show a bright letter "C" (Fig. 6B and Supplementary Movie 6). We also fabricated a bidirectional bending actuator by assembling two uni-directional bending segments (Fig. 6C and Supplementary Fig. 20B, D). Before the actuation, the actuator was invisible in the dark (Fig. 6D, Left). Upon hydraulic actuation, two segments bent to opposite directions to show a bright letter "S" (Fig. 6D and Supplementary Movie 6). Finally, based on the soft robot in Fig. 5C, we connected each leg of the robot to a syringe reservoir of dinoflagellate culture solution and then actuated the four legs in different sequences (Fig. 6E–H and Supplementary Movie 7). In the dark, a smaller letter "I" was displayed when only one leg was actuated; a larger letter "I" was displayed when two legs in the diagonal were actuated; a letter "V" was displayed when two adjacent legs were actuated; and a letter "X" was displayed when all the four legs were actuated.

**An untethered magnetically controlled soft mechanoluminescent robot**. Remote and contactless control of soft devices and robots have been widely explored using light[61], ultrasound[62], and magnetic fields[63,64], which can greatly simplify the control and enable untethered systems. For our biohybrid mechanoluminescent device, we further demonstrate the remote activation through magnetic field-induced actuations.

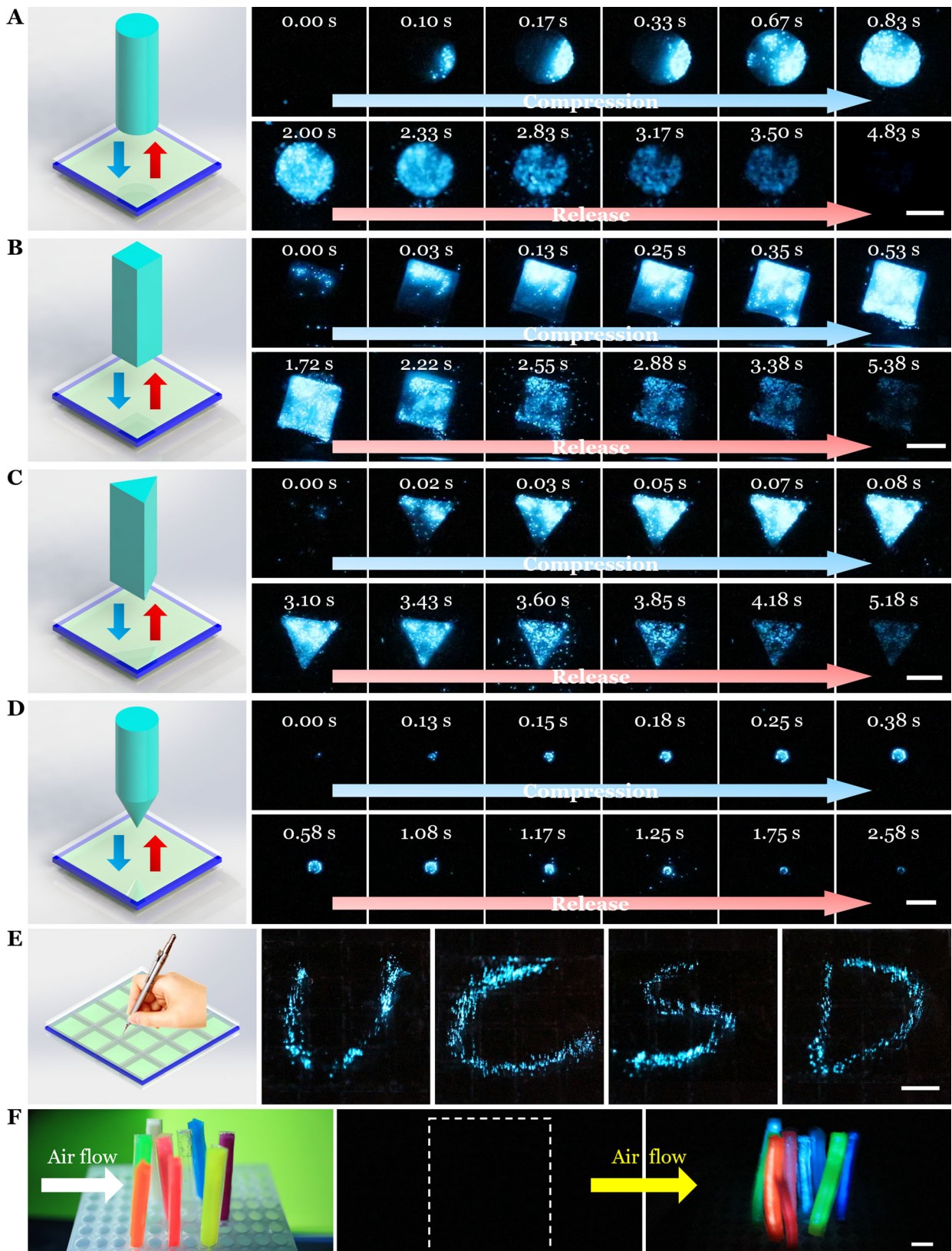

**Fig. 4 Activations of biohybrid mechanoluminescent devices by both contact and contactless stimuli to visualize external mechanical perturbations.** Bioluminescence, as viewed from below, was stimulated by contact of the biohybrid device with rigid objects including a (**A**) cylinder, (**B**) rectangular block, (**C**) triangular prism, and (**D**) cone. The objects were manually advanced until they contacted the biohybrid mechanoluminescent device to compress the dinoflagellates within; then the objects were released. Elapsed times are given in each frame. All scale bars are 5 mm. **E** Imaging of bioluminescence stimulated by writing the letters "UCSD" on the surface of a multichambered biohybrid device. Scale bar, 1 cm. **F** Photograph of colored cantilevers fixed to the base and bioluminescence from the colored biohybrid devices induced by air flow of ~45 L min⁻¹. Scale bar, 1 cm.

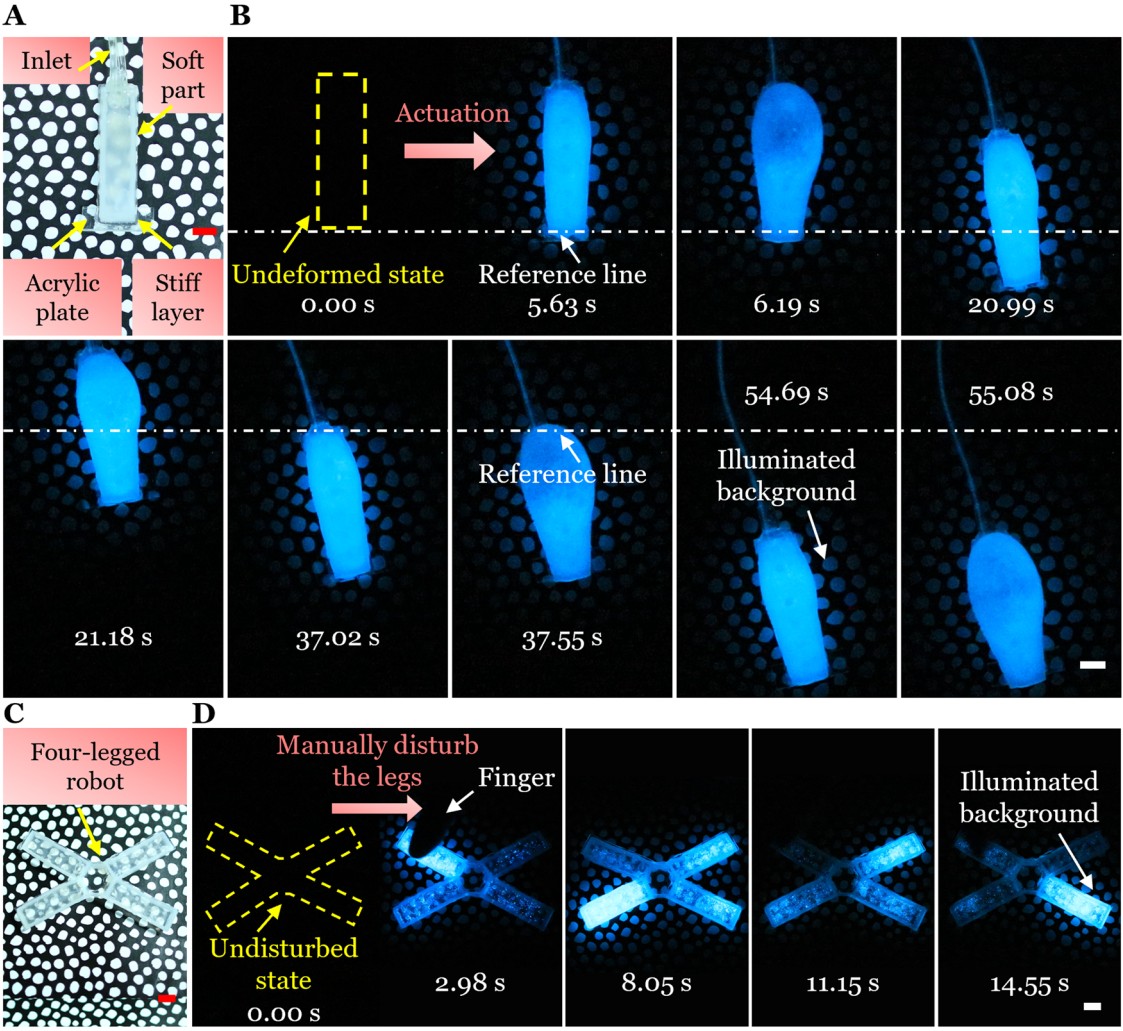

**Fig. 5 Active actuation-induced or passive mechanical perturbation-induced illumination of the biohybrid mechanoluminescent actuator/robot in the dark. A** A hydraulically actuated soft crawling robot. **B** Undeformed and actuated states of the soft robot in the dark at different times. Upon hydraulic actuation, the soft robot inflated and crawled forward and simultaneously produced light to make the surrounding background visible. **C** An untethered tetrapod-like soft robot with four legs. **D** Undisturbed and disturbed states of the four-legged robot in the dark at different times after mechanical disturbance. Pressed by a finger, the disturbed leg generated the brightest bioluminescence and illuminated the surrounding background while other legs produced much weaker bioluminescence. All scale bars are 1 cm.

To mimic bioluminescent marine animals[27,65], we built an untethered tetrapod-like soft robot that can be controlled by a remote magnetic stimulus. The design was modified from that in Fig. 5C, but permanent magnets were further introduced into each leg to enable magnetic activation. Specifically, the soft robot was composed of two parts: the tetrapod-like main body with a permanent magnet at the tip of each leg, and the transparent sealing layer (Fig. 7A and Supplementary Fig. 21). Depending on the position of the external activation magnet, the soft robot achieved diverse movement modes, resulting in fluid motion inside the legs that activated the bioluminescence (Fig. 7B and Supplementary Movie 8). When maintained in seawater under standard conditions for dinoflagellate cultures, light emission by the soft robot under magnetic actuation was maintained for at least 29 days (Fig. 7C and Supplementary Movie 9), indicating adequate oxygen and carbon dioxide supply for the dinoflagellates and also their high viability. Such mechanoluminescence correlated with actuation may functionalize untethered soft robots with sensing or illumination capabilities that can be maintained much longer than most previous biohybrid devices.

## Discussion

Biohybrid is a promising approach to construct highly energy efficient, autonomous, and biomimetic soft machines and robotics. Previous studies have clearly demonstrated the feasibility of integrating biological organisms with engineering materials or structures to enable new functions[1–4]. However, the limitations of most previously developed biohybrid devices have also been widely recognized, which include relatively short lifetimes typically within several days without careful maintenance[3,7–10,12], requirements of special maintenance to keep their functions and prolong lifetimes (e.g., well-controlled temperature and humidity, immersion in culture medium)[3,7–10,12], complex fabrication process[66], and slow responses (e.g., several hours for *E. coli* based biohybrid sensors)[7–9]. In the current work, we developed a simple method to fabricate highly robust, power-free, and soft biohybrid mechanoluminescent devices by integrating dinoflagellates with elastomer chambers. Diverse mechanical stimuli can trigger bioluminescence owing to the ultra-high force sensitivity of dinoflagellates, with rapid response time. The color range of the device can be extended by mixing color dyes into the elastomers. The

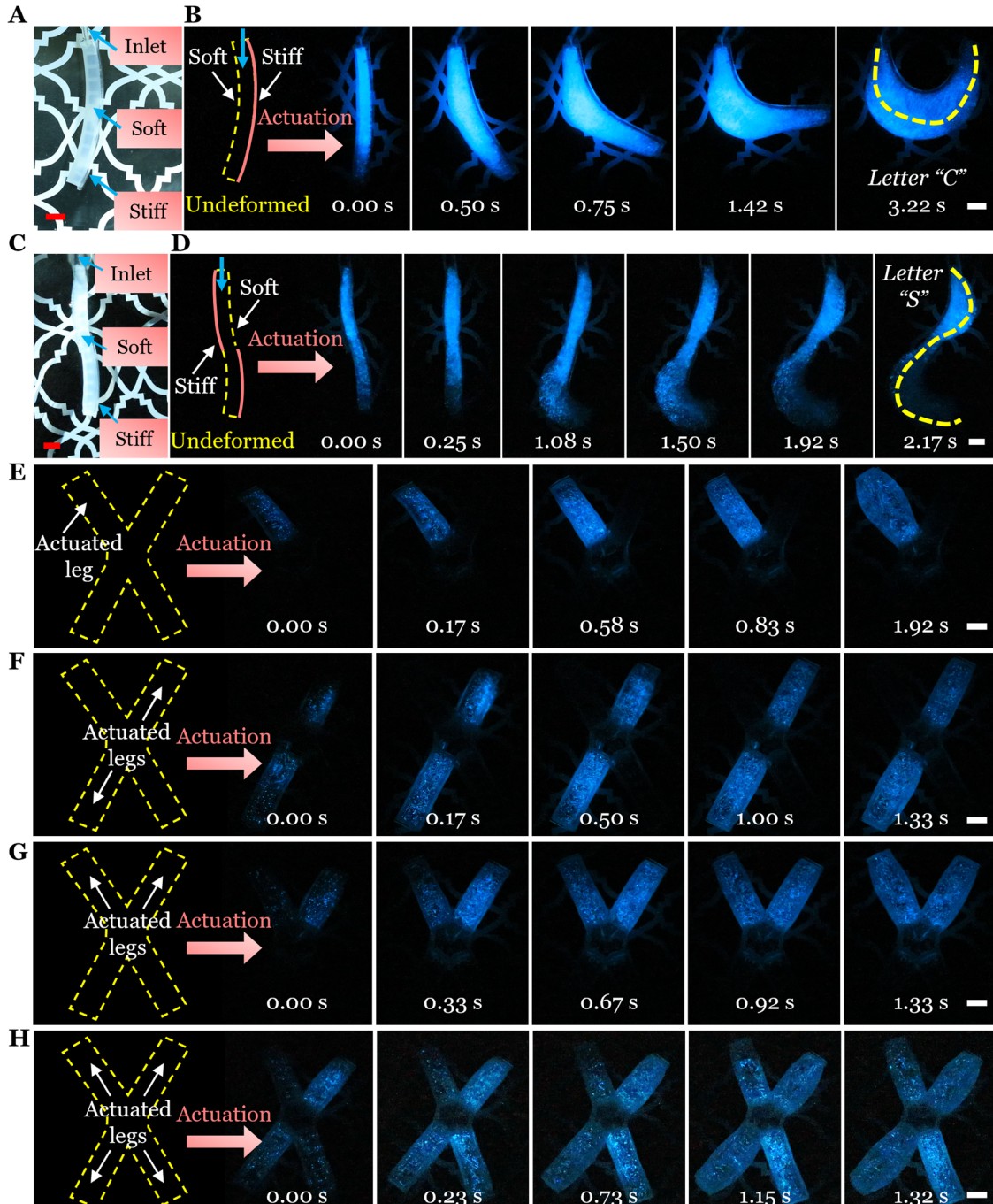

**Fig. 6 Optical signaling by the actuation of the biohybrid mechanoluminescent actuator/robot in the dark. A** Soft uni-directional bending actuator. **B** Actuation-induced optical signaling of the actuator in the dark. The actuator was composed of a soft inflatable part and a stiff constraining layer. Upon hydraulic actuation, the soft actuator bent to show a bright letter "C". **C** Soft bidirectional bending actuator. **D** Actuation-induced optical signaling of the actuator in the dark. The actuator was constructed by connecting two uni-directional bending actuators shown in A. Upon hydraulic actuation, two segments bent to opposite directions to show a bright letter "S". **E**–**H** Actuations of the four legs of a soft robot in different sequences to show different bright patterns for optical signaling. All scale bars are 1 cm.

light intensity of the device can also be enhanced by introducing small pillars into the chambers. We also demonstrated potential applications of these mechanoluminescent devices for soft robotics such as visualizing external mechanical perturbations, illuminating the surroundings, and optical signaling in the dark environment. An untethered magnetically controlled soft robot was further developed with a closed system and a lifetime of at least four weeks.

The current study has some limitations that can be further improved. First, because the dinoflagellate culture solution is encapsulated in an elastomer chamber, it is difficult to scale down the device to small sizes (~mm) and fabricate complex structures. A soft biohybrid composite may be more practical for applications. One feasible way to fabricate such composites is to embed dinoflagellate cells into a soft and biocompatible hydrogel matrix. Similar methods have been adopted to embed cells, bacteria, and

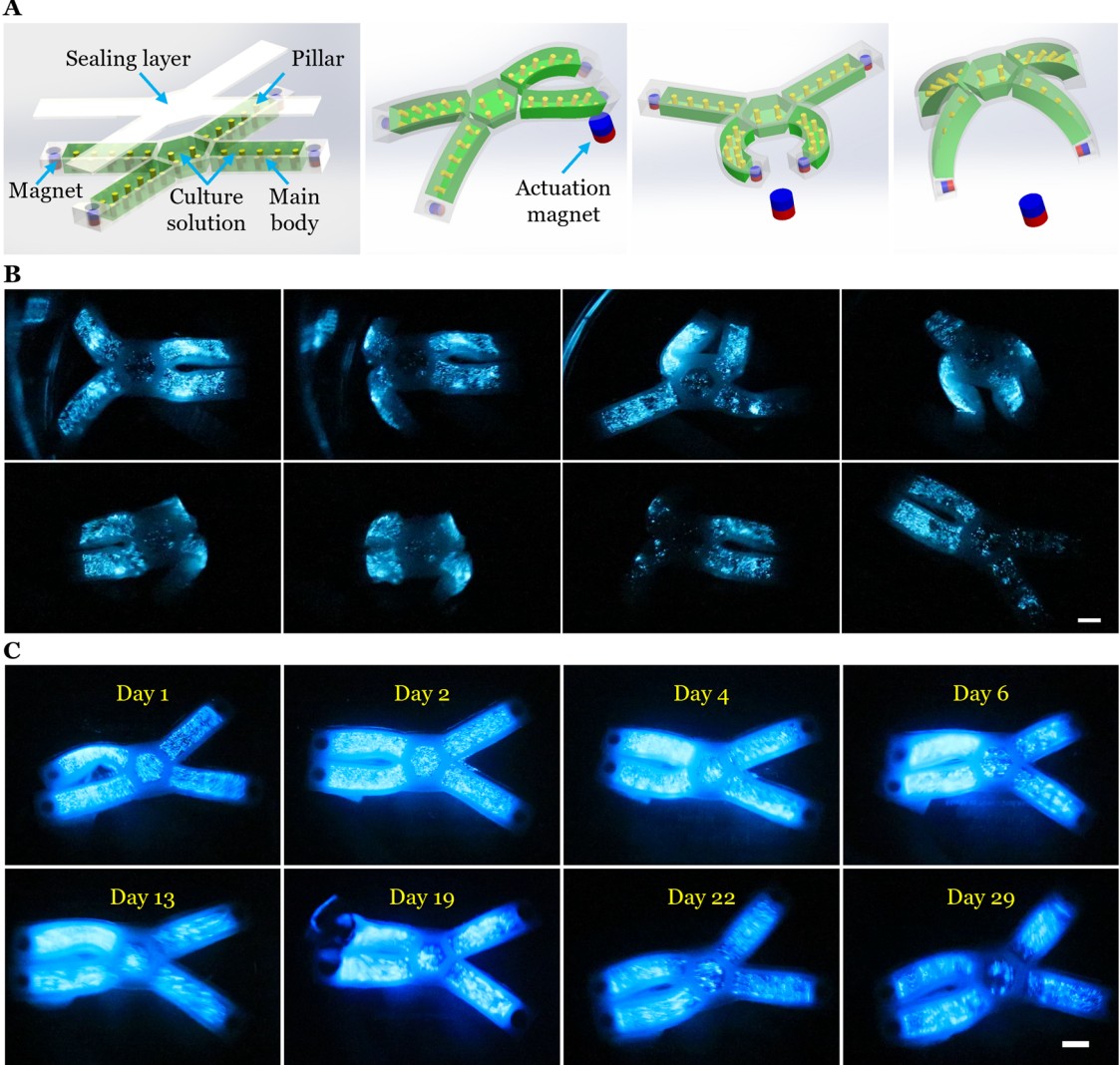

**Fig. 7 An untethered magnetically controlled soft mechanoluminescent robot with a closed system. A** Schematic of an untethered tetrapod-like soft robot that was controlled by a remote magnetic stimulus. Permanent magnets were embedded at the end of each leg, which consisted of a chamber with pillars. A transparent sealing layer covered the entire body of the robot. The dinoflagellate culture solution was injected into each leg. The motion of the robot was controlled by placement of an actuation permanent magnet. The motion and deformation of the soft robot activated the bioluminescence. **B** Bioluminescence stimulated by motion of the soft robot under magnetic actuation in seawater. The soft robot achieved varied luminescent patterns depending on the magnetic field-induced deformation mode. Scale bar, 1 cm. **C** Maintenance of bioluminescence of the magnetically controlled biohybrid soft robot. Bioluminescence was stimulated by motion of the soft robot under external magnetic actuation in seawater for at least 29 continuous days. Scale bar, 1 cm.

microalgae in hydrogels for functional living devices[9,10,67,68]. Such a dinoflagellate-hydrogel composite can be also used for detecting certain chemical changes in a fluid. For example, the change of pH value in a solution can activate the embedded dinoflagellates to emit light[34,69]. Second, for our current biohybrid devices, the mechanical deformation is transduced to bioluminescence, which is then detected optically. For the application of sensors, electrical signals may be much easier to be processed and analyzed. Therefore, it may be necessary to convert the light to electrical signals in certain applications, e.g., by embedding a photodiode or low light sensors into the device, similar to the design in a previous study[7], or by using a more sensitive solid-state photomultiplier. Third, the time duration for continuous light emission from the device is limited since the bioluminescent process consumes the finite amount of luminescent substrate in the cells. In the current study, the full recovery of the bioluminescent signal took up to 30 min after the complete consumption of the luminescent substrate. During the recovery period, refilling the biohybrid device with fresh culture solution from a reservoir is a feasible way to overcome this limitation. Fourth is the need to establish a quantitative relationship between mechanical stimuli and light intensity, which can be possibly achieved by using a solid dinoflagellate-hydrogel composite instead of liquid filled chambers, given that a quantitative model has been proposed for flow stimulated bioluminescence of dinoflagellates[70] and a phenomenological model has also been formulated for quantifying the single-cell bioluminescence[33]. Finally, dinoflagellates cannot tolerate extreme environmental conditions; *P. lunula* used in the current study can be maintained at temperatures between 18 to 27 °C[71–73]. Similar limitations are probably shared by most biohybrid devices.

Overall, as an initial approach to integrate bioluminescence with soft devices and robotics, we have introduced a simple method to fabricate highly robust and power-free soft biohybrid

mechanoluminescent devices. We have also explored the potential applications of mechanoluminescence for soft robotics in dark conditions. Most notably, our biohybrid mechanoluminescent devices are ultra-sensitive with fast response time and can maintain their light emission capability for at least four weeks without special maintenance. We believe that introducing intelligent biological behaviors into soft robotics will be an important future goal to enable more novel applications.

## Methods

**Culturing of dinoflagellates**. Cultures of the photosynthetic dinoflagellate *P. lunula* Schütt[74] were grown at 20 °C in half strength f/2 culture medium minus silicate on a 12:12 h light:dark cycle at the Scripps Institution of Oceanography. Prior to tests, we illuminated cultures with a 10 W LED lamp (Utilitech Pro E360557) from 0700–1900 h local time for the "light phase" to allow photosynthesis to occur to provide energy to the organisms, with a "dark phase" from 1900–0700 h during which time the bioluminescence system is activated[75,76]. Every other week, fresh culture medium was added in a 1:2 volume ratio. Each *P. lunula* cell is approximately 40 μm wide and 130 μm long (Supplementary Fig. 1B). Cell stiffness, based on Young's modulus as measured using atomic force microscopy, is 0.56 MPa[32].

**Fabrication of the biohybrid devices**. We first printed a polylactic acid (PLA) mold designed by Autodesk Inventor 2016 software with the 3D printer (Flash-Forge Guider 2) (Supplementary Fig. 2). Unless otherwise specified, we prepared the PDMS precursor with the same procedures. In brief, we mixed the base and curing agent of Sylgard 184 (Dow Corning) at 25:1 weight ratio, and then added a Pt-catalyst (Gelest SIP6831.2) at 0.5 uL g$^{-1}$ to accelerate the curing. The PDMS precursor was degassed with a vacuum pump for 5 min, and then poured into the mold for curing at room temperature. Simultaneously, extra PDMS precursor was poured into a laser cut polymethyl methacrylate (PMMA) mold to make the transparent sealing layer. After curing, we punched two holes on one sealing layer. Then, the molded PDMS structure and the sealing layer were glued together with the PDMS precursor. The geometry and dimension of the different molds used in this study are specified in detail in Supplementary Figures.

Toward the end of the light phase, when the bioluminescence system of *P. lunula* is not mechanically stimulable[75], the dinoflagellate culture solution was gently mixed to homogenize the distribution of cells, and then transferred into a plastic syringe and then injected into the PDMS chamber through the holes. Bubbles were eliminated by repeating this injection process until most bubbles escaped from the elastomer chamber. Next, we used the PDMS precursor to seal the holes. The injection of culture solution and sealing of holes were completed prior to the transition to the dark phase, when the biohybrid device was maintained in darkness. Note that the culture medium was not refreshed in the current study once the biohybrid device was fabricated to form a closed system. Unless otherwise specified, all characterizations and demonstrations of the biohybrid devices were conducted 4 h after the transition to the dark phase when bioluminescence is maximally expressed[75].

**Transmittance measurements**. Optical transmission spectra of pure PDMS and the biohybrid device was measured using UV–Vis spectroscopy (UNICO 4802 UV/Vis double beam spectrophotometer) (Supplementary Fig. 3).

**Uniaxial tension of the biohybrid device**. The uniaxial tension of the biohybrid device was conducted by a tensile machine (5965 Dual Column Testing Systems; Instron) with a 5000 N load cell and loading speed of 0.25 mm s$^{-1}$ (Supplementary Fig. 4).

**Mechanoluminescence imaging**. Unless otherwise specified, all photos and videos in this study were taken by a digital camera (Canon 80D) fitted with a Canon EF 24 mm f/1.4 L II USM wide angle lens. The aperture was always set to f/1.4 to allow maximum light collection. In the picture mode of the camera, ISO and exposure time were adjusted to capture the bioluminescence. In the video mode of the camera, ISO was adjusted to capture the bioluminescence while the video frame rate was always 60 fps.

**Vibration of the mechanoluminescent cantilever beam**. The PDMS chamber with cylindrical pillars was fabricated (Supplementary Fig. 5) and then infused with dinoflagellate culture solution. The device was clamped at the left end with an iron support. During the dark phase, we manually bent the right tip of cantilever to a certain degree and then suddenly released it, allowing the cantilever to freely vibrate in the vertical direction. We recorded the vibration process along with light emitting with the video mode of the camera at an ISO of 12,800.

**Extension of the color range**. During the fabrication process, we added Ignite fluorescent pigments (Smooth-On) into the PDMS precursor in a 1% weight:weight ratio (Supplementary Fig. 6). The cantilever was then vibrated as previously

described. We recorded the vibration process with the video mode of the camera at an ISO of 12,800.

**Characterizations of light intensity of the biohybrid devices**. All tests in this section were performed with the Electroforce 3300 mechanical test instrument (TA Instruments 3330-ES Series 3). Two kinds of chamber designs were used (Supplementary Figs. 4A and 5). A triangle loading with the maximum displacement as $\Delta x$ and the period as $\Delta t$ was applied to the samples. The maximum strain and strain rate were calculated as $\Delta x/l_0 \times 100\%$ and $2\Delta x/(l_0 \times \Delta t) \times 100\%$, respectively, where $l_0$ was the initial length of the chamber. Values of light emission represent averages with standard deviations for at least $N = 3$ replicates.

During each cycle of the test, we set the camera with a longer exposure time than the period to obtain an image. The exposure time was 5 s for the tests of pillar height (Fig. 3B and Supplementary Fig. 9), 6 s for the effect of strain rate (Fig. 3C and Supplementary Fig. 10A), 4 s for all the tests in Fig. 3D, Supplementary Fig. 8, 10B, 11B, 12–14, and 8 s for Supplementary Fig. 11A. The ISO was always set as 8000. We calculated the light intensity in grayscale with arbitrary units (arb. units) based on MATLAB image processing as follows (Supplementary Fig. 7): the original RGB image (8-bit for each channel) was first converted to an 8-bit grayscale format, and then an area outlining the boundary of the sample was selected. The light intensity was defined as the summation of the grayscale values divided by the number of pixels within the selected area. Similar definitions of light intensity have been used in previous studies[33,77]. A limitation of quantifying light emission from color images is that when the light intensity is high enough, some pixels of the original RGB image may be saturated in the blue channel, with an 8-bit value of 255. Thus, the light intensity determined through grayscale analysis was an underestimate of the actual light intensity.

To study the effect of the concentration of dinoflagellate cells on light intensity (Supplementary Fig. 8), chambers with $h = 2.5$ mm pillars were infused with various concentrations of dinoflagellate cultures during the light phase and then tested during the dark phase with the maximum strain and strain rate as 50 and 80% s$^{-1}$, respectively. On the same day, three regions of each device, each 30 mm$^2$ in area, were observed under the microscope. We then counted the number of cells in each image and calculated the average concentration of cells with standard deviations. The amount of blue channel pixel saturation was related to the cell concentration (Supplementary Fig. 8F).

For tests in Fig. 3C, D and Supplementary Figs. 12–14, chambers with $h = 2.5$ mm pillars were used. For tests in Fig. 3C and Supplementary Fig. 10A, the strain rate was varied while the maximum strain was kept as 50%. For tests in Fig. 3D and Supplementary Fig. 10B, the maximum strain was varied while the strain rate was kept as 50% s$^{-1}$. Similarly, we conducted the same experiments for the chamber without pillars (Supplementary Fig. 11). The effect of repeated cycles of stretching (Supplementary Fig. 12A, C) and compression (Supplementary Fig. 12B, D) were tested. To investigate the effect of elapsed time on the recovery of light emission (Supplementary Fig. 13), we first applied one cycle of the triangle test to the device and recorded the light intensity as $I_0$. Then, after a range of elapsed times, we repeated the same test and recorded the light intensity as $I_1$. The recovery ability is defined as the relative light intensity $I_1/I_0$.

For the viability tests (Supplementary Fig. 14), replicate chambers with $h = 2.5$ mm pillars were loaded with dinoflagellate cultures on day 1. The concentration of cells inside the chambers was measured on days 1, 4, 8, 11 and 15 as previously described. During each dark phase, we applied the same loading to all the devices over 15 continuous days with the maximum strain and strain rate as $-50$ and 80% s$^{-1}$, respectively. The stimulated bioluminescence was simultaneously imaged with the camera (Supplementary Fig. 14A). In between tests, the biohybrid devices were immersed in seawater and maintained on a 12:12 h light:dark cycle. The relative light intensity was defined as $I_i/I_{1st}$, where $I_{1st}$ and $I_i$ were the light intensity of the first day and subsequent day $i$, respectively. Note that the culture medium was not refreshed in this viability test as described in the section of fabrication of the biohybrid devices.

**Visualizing external mechanical perturbations in the dark**. We first printed objects with various geometries including a cylinder, rectangular block, triangular prism, and cone and then fabricated the single chamber biohybrid device (Supplementary Fig. 15). During the dark phase, we manually handled the object to compress onto and then release from the biohybrid device. The compression direction was perpendicular to the surface of the biohybrid device. The camera was set to video mode with an ISO of 12,800.

The multichambered panel was fabricated (Supplementary Fig. 16) and then infused with dinoflagellate culture solution. During the dark phase, we used a stylus to write "UCSD" letters on the top surface. Camera settings were ISO 6400 and exposure time of 8 s.

For the air flow test, multiple colored slender cantilevers were fabricated (Supplementary Fig. 17) and then fixed to a base. Then, an air flow of ~45 L min$^{-1}$ was applied across the biohybrid device. The camera was set to video mode with an ISO of 12,800.

**Illumination by actuations or disturbances in the dark**. The soft crawling robot was fabricated by assembling a soft inflatable part (Ecoflex-0030) and a stiff

constraining layer (PDMS with 10:1 ratio in weight) (Supplementary Fig. 18). A piece of acrylic plate was attached to the front of the robot to induce friction difference on the substrate. Two tubes were connected to the robot for infusion and actuation purpose. The robot was first filled with dinoflagellate culture solution and then one tube was blocked. The other actuation tube was connected to a syringe reservoir of dinoflagellate culture solution. In the dark phase, the robot was placed on an incline with the angle of 17º that had a patterned background. Then, we cyclically pushed the syringe to inject the solution into the robot and then pulled the syringe to extract the solution from the robot. As a result, the soft robot was inflated and deflated repeatably and simultaneously crawled forward.

For the external disturbance-induced illumination, we designed an untethered tetrapod-like soft robot and then filled the four legs with dinoflagellate culture solution (Supplementary Fig. 19). PDMS precursor was then used to seal the holes on four legs. In the dark phase, the four-legged robot was put on a patterned background. Then, we manually disturbed each leg by pressing.

The camera was set to video mode with an ISO of 12,800 to record the above processes.

**Optical signaling by actuations in the dark**. We firstly designed a hydraulically actuated soft bending actuator (Supplementary Fig. 20A, C). The uni-directional bending actuator was composed of a soft inflatable part and a stiff constraining layer. Two tubes were connected to the actuator for infusion and actuation purpose. The actuator was first filled with dinoflagellate culture solution and then one tube was blocked. The other tube was connected to a syringe reservoir of dinoflagellate culture solution for hydraulic actuation. In the dark phase, we placed the actuator in a water bath to reduce the effect of gravity. Upon hydraulic actuation, the soft actuator bent uni-directionally.

The bidirectional bending actuator was fabricated by assembling two uni-directional bending segments in an alternating sequence (Supplementary Fig. 20B, D). Upon hydraulic actuation, two segments bent to opposite directions.

For the four-legged robot, we connected each leg of the robot to a syringe reservoir of dinoflagellate culture solution and then actuated the four legs in different sequences. Thus, different letters or patterns could be displayed.

The camera was set to video mode with an ISO of 12,800 to record the above processes. For the bioluminescence images presented (Figs. 6D, E–H), the original frames extracted from the video were increased in brightness to 115 and 150%, respectively.

**An untethered magnetically controlled soft mechanoluminescent robot**. The main body of the untethered soft robot was made of Ecoflex-0030 (Smooth-On) (Supplementary Fig. 21). In brief, Part A and part B of Ecoflex-0030 were mixed with 1:1 weight ratio. After degassed for 5 min, the precursor was poured into the 3D printed mold to fabricate the main body part. Similarly, PDMS precursor was poured into a PMMA mold to make a transparent sealing layer. After curing, permanent magnets were embedded into predesigned cavities of four legs of the soft robot. Then we sealed the cavities with the precursor of Ecoflex-0030. The sealing layer and the main body were assembled together, and then dinoflagellate culture solution was injected into the soft robot. For tests during the dark phase, the soft robot was placed in a tank containing seawater and an actuation magnet was manually handled to control the motion of the soft robot. For repeated tests over 29 consecutive nights, the soft robot was maintained in seawater on a 12:12 h light:dark cycle. Note that the culture medium was not refreshed in this repeated test of the robot as described in the section of fabrication of the biohybrid devices. Bioluminescence was imaged with the camera set to video mode with an ISO of 12,800.

## Data availability

The authors declare that the data supporting the findings of this study are available within the paper and its Supplementary Information files or from the corresponding authors on reasonable request.

## Code availability

The custom codes used in this study for video processing and light intensity calculation are available at https://github.com/ChenghaiLi/MATLAB-Code.

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

## Acknowledgements

The authors acknowledge support from the Office of Naval Research through grant no. N00014-17-1-2062 and the US Army Research Office through grant no. W911NF-20-2-0182. We thank R. Chen and T. Feng for assisting with the UV–Vis spectroscopy by measuring the transmittance of the biohybrid devices.

## Author contributions

C.L., M.I.L., and S.C. jointly conceived the soft biohybrid mechanoluminescence concepts and designed the study. C.L. developed materials and methods of fabrication, performed devices characterizations, and completed the luminescent demonstrations. Q.H., Y.W., Zhijian Wang, Zijun Wang, R.A., M.I.L., and S.C. provided suggestions on designing experiments and analyzing the results. C.L., M.I.L., and S.C. wrote the manuscript with input from all authors. M.I.L. and S.C. supervised the study.

## Competing interests

The authors declare no competing interests.
