## [Peer Review File · Nature Communications]

Highly robust and soft biohybrid mechanoluminescence for optical signaling and illuminationREVIEWER COMMENTS

Reviewer #1 (Remarks to the Author):

In this study, the authors encapsulated bioluminescent unicellular marine algae, dinoflagellates, into soft elastomeric chambers to generate highly robust and power-free biohybrid mechanoluminescence. This biohybrid mechanoluminescent device have been proved to be ultra-sensitive with fast response time, and could maintain their light emission capability for a while. Additionally, the biohybrid mechanoluminescent device could be made into various geometries for application in different fields, such as visualizing external mechanical perturbations, deformation-induced illumination, and optical signaling in dark environment. The design and preparation of such devices seems simple. But there are still some major concerns that hinder the publication of this work.

1. The culture medium containing dinoflagellates was injected into the closed PDMS chamber. How to ensure the oxygen required for the long-term life activities of dinoflagellates?
2. Dinoflagellates require high living conditions. The device cannot guarantee the use under extreme conditions, such as high temperature, low temperature, etc.
3. As far as I know, bioluminescence means energy consumption of organisms. How to ensure that the culture medium provides sufficient nutrition for the dinoflagellates to realize their long-term continuous illumination under dark conditions? What is the maximum duration of continuous illumination of the device under current conditions?
4. Does the R2 value of only 0.35 in Supplementary Figure 14 mean that the survival rate of dinoflagellates is too low?

Reviewer #2 (Remarks to the Author):

The work developed a simple method to fabricate highly robust, power-free, and soft biohybrid mechanoluminescent devices by integrating dinoflagellates with elastomer chambers. This biohybrid mechanoluminescent devices demonstrate potential applications such as visualizing external mechanical perturbations, deformation-induced illumination, and also optical signaling in dark environment. This work is very interesting and significant.

- 1 Briefly demonstrate the process of fabrication of the biohybrid device and how to avoid bubbles?
- 2 How can the intensity of light be quantified to determine the magnitude of mechanical forces?
- 3 How is the culture medium changed during the biohybrid device recycling process? How many days does it take to change the culture medium once?
- 4 What is advantage of this method compared with existing based methods? I think "10.1126/scirobotics.aar8580; Proc. Natl. Acad. Sci. U.S.A. 2020, 117, 22736-22742; 10.1016/j.nantod.2021.101268 are useful to you.
- 5
、 The author should explore the biohybrid device whether can make some response changes in the complex biological fluid (such as the change of some ion concentration in blood), or capture some biological samples under the manipulation of magnetic field?

Comments from reviewer 1:

In this study, the authors encapsulated bioluminescent unicellular marine algae, dinoflagellates, into soft elastomeric chambers to generate highly robust and power-free biohybrid mechanoluminescence. This biohybrid mechanoluminescent device has been proved to be ultra-sensitive with fast response time and could maintain their light emission capability for a while. Additionally, the biohybrid mechanoluminescent device could be made into various geometries for application in different fields, such as visualizing external mechanical perturbations, deformation-induced illumination, and optical signalling in dark environment. The design and preparation of such devices seems simple. But there are still some major concerns that hinder the publication of this work.

Response to reviewer 1:

1. Q: *The culture medium containing dinoflagellates was injected into the closed PDMS chamber. How to ensure the oxygen required for the long-term life activities of dinoflagellates?*

A: We thank the reviewer for carefully evaluating our manuscript as well as proposing valuable comments. In addressing this comment, we need to point out that the dinoflagellate *Pyrocystis lunula* is photosynthetic. Therefore, cells require carbon dioxide for growth via photosynthesis. In our study, we were extremely careful to ensure the survivability of the dinoflagellates within the devices:

(1) We used PDMS to fabricate the elastomer chambers. PDMS is known to be highly permeable to gas exchange (Liu et al., 2017; Robb, 1968; Stern, 1994; Aoki, 1999). Dinoflagellates of the genus *Pyrocystis* are known to tolerate being enclosed within containers (Latz, 2017).

(2) For our biohybrid devices, the dinoflagellates inside the PDMS chamber experienced a 12:12 h light:dark physiological cycle (Stauber et al., 2008). During the light phase, the transparent PDMS allowed the transmission of light and dinoflagellates carried out photosynthesis to produce oxygen, which was then respired during the dark phase.

(3) As shown in **Supplementary Figure 14** in the revised manuscript, and described in **Supplementary Note 4**, the dinoflagellates grew and survived well in our biohybrid devices. After an initial period with some cell mortality, the average cell concentration from day 4 through day 11 increased at an exponential rate that was consistent with previous studies of dinoflagellate growth in liquid cultures with air exchange (Sullivan and Swift, 2003; Latz et al., 2009). Furthermore, the light intensity of the device on the 15th day was similar to that on the 2nd day, which indicated the stability of the system.

(4) Moreover, as shown in **Figure 7C**, the magnetically controlled biohybrid robot (a closed environment for dinoflagellates) maintained its light emission functions for at least 29 days, which was the end of the experiment.

In summary, there was no indication that dinoflagellates enclosed in the biohybrid devices experienced limitation due to gas exchange. To clarify this point, we modified/added additional text and related references to the revised manuscript:

Page 3 3rd paragraph: “In the light phase, the soft robot integrated with dinoflagellate culture solution is charged with light for photosynthesis to produce oxygen, providing energy for the organism.”

Page 4 1st paragraph: “In addition, dinoflagellates of the genus *Pyrocystis* are known to tolerate being enclosed within containers⁵³. Note that PDMS is highly permeable to gas exchange^{8, 54-56}, which guarantees the survival of enclosed bacteria/cells.”

Page 4 1st paragraph: “The biohybrid device is highly transparent to visible light (Fig. 2B, Left and Supplementary Fig. 3), allowing both the transmission of illumination for the dinoflagellates to carry out photosynthesis to produce oxygen during the light phase”

Page 5 3rd paragraph: “Our biohybrid devices could maintain a high relative light intensity (>55% of initial values) for at least 15 days (Supplementary Fig. 14B), indicating adequate oxygen and carbon dioxide supply for the dinoflagellates and also high viability.”

Page 7 3rd paragraph: “When maintained in seawater under standard conditions for dinoflagellate cultures, light emission by the soft robot under magnetic actuation was maintained for at least 29 days (Fig. 7C and Supplementary Movie 9), indicating adequate oxygen and carbon dioxide supply for the dinoflagellates and also their high viability.”

Page 20 caption for Figure 1: “In the light phase, the soft biohybrid robot integrated with dinoflagellate culture solution is charged with sunlight for photosynthesis to produce oxygen, providing energy for the organism.”

2. Q: *Dinoflagellates require high living conditions. The device cannot guarantee the use under extreme conditions, such as high temperature, low temperature, etc.*

A: In nature, the dinoflagellate *Pyrocystis lunula* has a very wide global distribution, being found in the Atlantic, Pacific, and Indian Oceans, as well as the Mediterranean and China Seas. Therefore, it naturally experiences a wide range of environmental conditions. In the laboratory, it can be grown at temperatures

of 18 to 27°C. This temperature range is suitable for many applications. So, we agree with the reviewer that dinoflagellates are not suitable for extreme conditions. However, most biohybrid devices are unable to maintain functionality under extreme conditions, an unavoidable limitation of using biohybrid devices.

To clarify this, we modified/added some text in the revised manuscript:

Page 3 last paragraph: “This species was selected because it is widespread globally in many oceans of the world⁴⁸⁻⁵², indicating that it is tolerant to a broad range of environmental conditions.”

Page 8 1st paragraph: “Finally, dinoflagellates cannot tolerate extreme environmental conditions; *P. lunula* used in the current study can be maintained at temperatures between 18 to 27°C⁷¹⁻⁷³. Similar limitations are probably shared by most biohybrid devices.”

3. Q: *As far as I know, bioluminescence means energy consumption of organisms. How to ensure that the culture medium provides sufficient nutrition for the dinoflagellates to realize their long-term continuous illumination under dark conditions? What is the maximum duration of continuous illumination of the device under current conditions?*

A: The dinoflagellate *Pyrocystis lunula* requires light conditions for photosynthesis and growth, and dark conditions to express bioluminescence. In this study, we demonstrated that dinoflagellate biohybrid devices can maintain their functionality for weeks when maintained on a light and dark cycle. This is a major advantage over the limited life span of many other biohybrid devices. Furthermore, light emission can recover from repeated stimulation (**Figure 2, Supplementary Movie 1, and Supplementary Figure 12**).

Importantly, as shown in **Supplementary Figure 13**, there is recovery of light emission when the biohybrid devices are left in a dark environment. Relative light intensity increased with the elapsed time for both stretching and compression, consistent with the recovery of bioluminescence measured for *P. fusiformis*, which takes 30 min for full recovery (Widder and Case, 1981). Thus, although the light intensity keeps decreasing under the same mechanical stimulus, a short time of 30 min is enough for our biohybrid devices to fully recover its light emission capability in the single dark phase.

To clarify this, we wrote the following sentences in the **Discussion** section on Page 8 in the original manuscript. To be clear, it was also marked in red in the revised manuscript:

Page 8 1st paragraph: “Third, the time duration for continuous light emission from the device is limited since the bioluminescent process consumes the finite amount of luminescent substrate in the cells. In the current study, the full recovery of the bioluminescent signal took up to 30 min after the complete consumption of the luminescent substrate. During the recovery period, refilling the biohybrid device with fresh culture solution from a reservoir is a feasible way to overcome this limitation.”

4. **Q:** Does the R^2 value of only 0.35 in **Supplementary Figure 14** mean that the survival rate of dinoflagellates is too low?

A: In **Supplementary Figure 14F**, the blue line represented the linear regression between the saturation ratio in the blue channel of the RGB image and the cell concentration up to 3.9 cells mm^{-2} with $R^2 = 0.35$; this is a characteristic of the imaging system and has nothing to do with cell viability. Therefore, **Supplementary Figure 14F** does not imply that the survival rate of dinoflagellates inside the elastomer chambers is low.

Instead, **Supplementary Figure 14B** and **C** quantitatively showed the changes of light intensity and cell concentration for 15 continuous days, which indicated high survival rate of dinoflagellates inside the elastomer chambers. In **Supplementary Figure 14B**, after an initial decrease in average relative light intensity for 9 devices to ~56% of initial intensity by day 3, light intensity was then maintained through the end of the experiment, indicating high viability of dinoflagellates inside the elastomer chambers.

Meanwhile, as shown in **Supplementary Figure 14C**, after an initial decline, average cell concentration increased exponentially at a rate that was consistent with previous studies of dinoflagellate growth (Sullivan and Swift, 2003; Latz et al., 2009). This result also indicated high viability of dinoflagellates inside the elastomer chambers.

Comments from reviewer 2:

The work developed a simple method to fabricate highly robust, power-free, and soft biohybrid mechanoluminescent devices by integrating dinoflagellates with elastomer chambers. This biohybrid mechanoluminescent devices demonstrate potential applications such as visualizing external mechanical perturbations, deformation-induced illumination, and also optical signalling in dark environment. This work is very interesting and significant.

Response to reviewer 2:

1. Q: *Briefly demonstrate the process of fabrication of the biohybrid device and how to avoid bubbles?*

A: We thank the reviewer for carefully evaluating our work and proposing constructive comments. The process of fabrication of the biohybrid devices and how to avoid bubbles are described as follows:

As shown in **Supplementary Figure 2** in the revised manuscript, we mixed the base and curing agent of Sylgard 184 (Dow Corning) at 25:1 weight ratio, and then added a Pt-catalyst (Gelest SIP6831.2) at 0.5 $\mu\text{L g}^{-1}$ to accelerate the curing. The PDMS precursor was degassed with a vacuum pump for 5 min, and then poured into the 3D printed polylactic acid (PLA) mold for fabricating the spacer layer. Simultaneously, extra PDMS precursor was poured into a laser cut polymethyl methacrylate (PMMA) mold to make the transparent sealing layer. After curing, we punched two holes on one sealing layer to allow the injection of dinoflagellate culture solution. Then, the spacer layer and two sealing layers were glued together with the PDMS precursor to form a PDMS chamber. The geometry and dimension of different molds used in this study are provided in detail in Supplementary Figures. Toward the end of the light phase, the dinoflagellate culture solution was gently mixed to homogenize the distribution of cells, and then transferred into a plastic syringe and injected into the PDMS chamber through the holes. Bubbles were eliminated by repeating this injection process until most bubbles escaped from the elastomer chamber. Next, we used the PDMS precursor to seal the holes. The injection of culture solution and sealing of holes were completed prior to the transition to the dark phase, when the biohybrid device was maintained in darkness.

To clarify this, we have further described the fabrication process both in the **Fabrication of the biohybrid devices** section on Page 9 and also in the caption for **Supplementary Figure 2** on Page 31.

2. Q: *How can the intensity of light be quantified to determine the magnitude of mechanical forces?*

A: Early studies related the intensity of light emission to properties of a fully characterized stimulating flow field (e.g., Latz et al., 1994; Latz and Rohr, 1999). Based on these and similar results, a 2005 study presented a statistic model for the flash response of bioluminescent dinoflagellates stimulated by fluid shear, based

on the idea that the response of an individual cell to stimulation is inherently probabilistic and can be modelled as a Poisson process over short time scales (Deane and Stokes, 2005). These experimental approaches are very challenging because the distribution of force stimuli across the individual cell is difficult to quantify. More recent studies using direct contact of restrained cells (Jalaal et al., 2020; Tesson and Latz, 2015) have demonstrated a viscoelastic response in which light intensity depends on both the amplitude and rate of deformation.

In the context of this study, we seek a quantitative relationship between light intensity and the magnitude of applied mechanical forces. However, this is very challenging due to the following reason. For our biohybrid devices, the external force deforms the elastomer chamber, inducing flow of the encapsulated culture solution, resulting in a shearing deformation of dinoflagellates and activating bioluminescence. The highly coupled process makes it hard to establish an explicit quantitative relationship between the light intensity and the applied mechanical forces. Therefore, for the design presented in the current study, it will be impossible to extract precise quantitative information of the applied forces from the light intensity.

Currently we have started a new study in which we embed dinoflagellates into a soft hydrogel matrix. This improved design will allow us to establish a quantitative relationship between light intensity and the magnitude of the applied force. For the solid dinoflagellate-hydrogel composite, the applied loading and loading rate can be precisely controlled, greatly simplifying the theoretical modelling.

We have expanded the discussion in the revised manuscript:

Page 8 1st paragraph: “**Fourth is the need to establish a quantitative relationship between mechanical stimuli and light intensity, which can be possibly achieved by using a solid dinoflagellate-hydrogel composite instead of liquid filled chambers, given that a quantitative model has been proposed for flow stimulated bioluminescence of dinoflagellates⁷⁰ and a phenomenological model has also been formulated for quantifying the single-cell bioluminescence³³.**”

3. Q: *How is the culture medium changed during the biohybrid device recycling process? How many days does it take to change the culture medium once?*

A: As shown in **Supplementary Figure 2**, once the dinoflagellate culture solution is injected into the PDMS chamber, the fabricated biohybrid device was a closed system. As shown in **Supplementary Figure 14** and described in **Supplementary Note 4**, our biohybrid devices showed relatively high light intensity under the same loadings after 15 continuous days without refreshing the culture medium.

In addition, as shown in **Figure 7C**, the magnetically controlled biohybrid robot with a closed system can maintain its light emission functions for at least 29 days without refreshing the culture medium. For repeated tests over the consecutive nights, the soft biohybrid device was maintained in seawater on a

12:12 h light:dark cycle. Both experiments (**Figure 7C** and **Supplementary Figure 14**) showed that dinoflagellates can survive for a long duration inside the closed PDMS chambers without refreshing the culture medium.

To clarify this, we modified/added the text:

Page 9, 3rd paragraph: “Note that the culture medium was not refreshed in the current study once the biohybrid device was fabricated to form a closed system.”

Page 11 2nd paragraph: “Note that the culture medium was not refreshed in this viability test as described in the section of fabrication of the biohybrid devices.”

Page 12 4th paragraph: “Note that the culture medium was not refreshed in this repeated test of the robot as described in the section of fabrication of the biohybrid devices.”

Page 31 caption of Supplementary Figure 2: “Note that the culture medium was not refreshed in the current study once the biohybrid device was fabricated to form a closed system.”

4. Q: *What is advantage of this method compared with existing based methods? I think*

“10.1126/scirobotics.aar8580; Proc. Natl. Acad. Sci. U.S.A. 2020, 117, 22736-22742; 10.1016/j.nantod.2021.101268 are useful to you.

A: The reviewer highlights pioneering studies in the field. All these references are related and inspiring, and should have been cited in our manuscript. We have added those papers into the reference list of the revised manuscript. The major difference between the current work and the previous studies is that photonic crystal array designs can change the structure color when subjected to mechanical deformations or change fluorescence intensity with the addition of corresponding target. However, they could not produce light, so they cannot be used in a dark environment. The biohybrid mechanoluminescent devices developed in current study can produce light under mechanical perturbations, which, therefore, can visualize deformations, illuminate surrounding area, and produce optical signals in dark environment.

To compare our work and previous studies, we modified/added the text and added related references in the revised manuscript:

Page 2 2nd paragraph: “Moreover, photonic crystal arrays have been explored to visualize mechanical deformations/forces¹⁶⁻¹⁸. However, none of these designs can emit light, so it is still very challenging to visualize deformation or forces in a dark environment.”

5. **Q:** *The author should explore the biohybrid device whether can make some response changes in the complex biological fluid (such as the change of some ion concentration in blood), or capture some biological samples under the manipulation of magnetic field?*

A: Though dinoflagellates can emit light with the change of ion concentration (Valiadi and Iglesias-Rodriguez, 2013; Fogel and Hastings, 1972), for the current design, the biohybrid device does not respond to the ion concentration change in solution because the PDMS skin is the barrier preventing the diffusion of ions into the dinoflagellate solution. As pointed out in the discussion section, we are exploring embedding dinoflagellates into hydrogel matrix to make a solid biohybrid composite, which can respond to the ion concentration change.

The primary goal of this study was to present the design principle of soft biohybrid mechanoluminescence. There is no intrinsic difficulty in combining the current design with other functionalities of soft robotic structures already demonstrated in previous studies, such as crawling and gripping. As an example, we have only demonstrated crawling mechanoluminescent robot in **Figure 5** and **Figure 7**. We indeed plan to explore more applications of the soft mechanoluminescent devices in the future studies.

In the revision, we have added the following text:

Page 7 last paragraph: “**One feasible way to fabricate such composites is to embed dinoflagellate cells into a soft and biocompatible hydrogel matrix.**”

Page 8 first paragraph: “**Such a dinoflagellate-hydrogel composite can be also used for detecting certain chemical changes in a fluid. For example, the change of pH value in a solution can activate the embedded dinoflagellates to emit light^{34, 69}.**”

Page 8 2nd paragraph: “**We believe that introducing intelligent biological behaviors into soft robotics will be an important future goal to enable more novel applications.**”

In addition to the comments from above two reviewers, we also modified some sentences in the revised manuscript for clarity:

Page 2 2nd paragraph: “**To produce light, a highly stretchable electroluminescent skin has been recently developed for soft robots for both optical signaling and tactile sensing¹³.**”

Page 6 last paragraph: “In the dark, a smaller letter “I” was displayed when only one leg was actuated; a larger letter “I” was displayed when two legs in the diagonal were actuated; a letter “V” was displayed when two adjacent legs were actuated; and a letter “X” was displayed when all the four legs were actuated.”

REVIEWERS' COMMENTS

Reviewer #1 (Remarks to the Author):

The authors had properly revised their manuscript and addressed my comments. I think the manuscript can be published in nature communication now.

Reviewer #2 (Remarks to the Author):

The authors have addressed all of my concerns.